# Stimulus selection enhances value-modulated somatosensory processing in the superior colliculus

**Yun Wen Chu[1,2☯], Suma Chinta[1☯], Hayagreev V. S. Keri[1,2☯], Shreya Beri[1], Scott R. Pluta[1]\***

**1** Department of Biological Sciences, Purdue University, West Lafayette, Indiana, United States of America
**2** Department of Biomedical Engineering, Purdue University, West Lafayette, Indiana, United States of America

☯ These authors contributed equally to this work.
* spluta@purdue.edu

## Abstract

A fundamental trait of intelligent behavior is the ability to respond selectively to stimuli with higher value. Where along the neural hierarchy does somatosensory processing transition from a map of stimulus location to a map of stimulus value? To address this question, we recorded single-unit activity from populations of neurons in somatosensory cortex (S1) and midbrain superior colliculus (SC) in mice conditioned to respond to a positive-valued stimulus and withhold responses to an adjacent, negative-valued stimulus. The stimulus preference of the S1 population was equally weighted towards either stimulus, in line with a somatotopic map. Surprisingly, we discovered a large population of SC neurons that were disproportionately biased towards the positive stimulus. This disproportionate bias was largely driven by enhanced spike suppression for the negative stimulus. Removing the opportunity for mice to behaviorally select the positive stimulus reduced positive stimulus bias and spontaneous firing rates in SC but not S1, suggesting that neural selectivity was augmented by task readiness. Similarly, the spontaneous firing rates of SC but not S1 neurons predicted reaction times, suggesting that SC neurons played a persistent role in perceptual decision-making. Taken together, these data indicate that the somatotopic map in S1 is transformed into a value-based map in SC that encodes stimulus priority.

## Introduction

Imagine that it is raining outside, and gusts of wind are ripping through the trees. While looking out the window, you approach the closet to find a jacket. You quickly dismiss a t-shirt before grabbing a parka and heading out the door. In daily life, we encounter countless objects in our environment. To optimize survival, we select the objects that add value to our life and dismiss the objects that add unnecessary costs. Guiding this behavior are maps of spatial features in our brain that are repeated across the different stages of sensory processing. Where along this hierarchy does information transition from a map faithful to spatial location to a map interwoven with stimulus value? To address this question, we utilized the somatosensory whisker system of mice, where a map of stimulus space is clearly organized across the different stages of sensory processing [1–8].

In this study, we focused on the primary somatosensory cortex (S1) and midbrain superior colliculus (SC), due to their known importance in sensory-guided behaviors [9–12]. Both

**Data availability statement:** Through Zenodo, all data and code is publicly available: https://zenodo.org/doi/10.5281/zenodo.13346093. https://zenodo.org/records/14790606.

**Funding:** This work was supported by the Whitehall Foundation (to SRP), Showalter Trust (to SRP), and Air Force Office of Scientific Research (FA9550-23-1-0701 to SRP). Any opinions, findings, and conclusion or recommendations expressed in this material are those of the author(s) and do not necessarily reflect the views of the United States Air Force. SC and YWC received a salary from the Whitehall Foundation and Showalter Trust. HVSK received a salary from AFOSR. The funders had no role in study design, data collection and analysis, decision to publish, or preparation of the manuscript.

**Competing interests:** The authors have declared that no competing interests exist.

**Abbreviations :** CR, Correct Rejection; DLC, DeepLabCut; ROI, region of interest; S1, somatosensory cortex; SC, superior colliculus.

brain areas receive bottom-up whisker information that encodes the location of stimuli in somatotopic coordinates. Both areas receive top-down input from motor and association cortices, providing a circuit basis for contextualizing stimulus information with goal-oriented actions [13–16]. S1 provides monosynaptic sensory drive to SC, and ascending SC projections to the thalamus augment whisker responses in S1, forming a reciprocal flow of tactile information [13,17,18]. Despite these known relationships, how sensory associations differentially shape their representations of space is unclear. While each brain area has been studied in isolation, there is no study comparing somatosensory processing in each area during the same goal-oriented task.

Behavioral context is known to modify neuronal receptive fields in S1 and SC. For example, while performing a shape discrimination task that involves multiple whiskers, S1 neurons located in the most task-relevant barrel column expand their receptive fields [19]. In addition, S1 neurons adapt their tactile sensitivity to match the intensity of the task-relevant stimulus distribution [20]. In SC, neuronal receptive fields expand and increase their sensitivity when the animal is prepared to make a saccadic eye movement [21,22], suggesting that movement preparation enhances SC visual processing. Therefore, receptive field properties in S1 may adapt to optimally encode the task-relevant stimulus features, while sensory processing in SC appears to have a stronger relationship to orienting movements and decision-making [23,24].

To determine how the representations of sensory space in S1 and SC are modulated by behavioral context, we recorded single-unit activity from populations of neurons in mice performing stimulus selection guided by value-based associations. In one version of the task, two adjacent single-whisker stimuli had opposite values (positive/negative) and elicited opposing behavioral responses (Go/No-Go). In the other version, the two adjacent single-whisker stimuli had equal value (positive) and elicited the same behavioral response (Go). In both versions of the task, the stimulus preference of the S1 population was equally weighted towards either whisker stimulus. Therefore, stimulus value and its associated action did not modify the representation of sensory space in S1. Conversely, we found that behavioral context strongly influenced the representation of whisker space in SC. SC neurons were strongly biased towards the positive valued stimulus, but only when the stimuli and their associated actions had opposite values. When the stimuli had equal value, stimulus bias in the SC population became symmetric and smaller than S1. Furthermore, removing the opportunity for mice to select the positive valued stimulus reduced bias in SC, but not S1. Likewise, the spontaneous firing rate of SC but not S1 neurons predicted the speed of perceptual decision-making. Taken together, these data suggest that S1 contains a faithful somatotopic map of whisker space that is transformed into a value-based and action-oriented map in SC that encodes stimulus priority.

## Results

To determine the impact of stimulus value on somatosensory processing in S1 and SC, we trained mice to discriminate between two adjacent single-whisker stimuli of opposing values. Mice performed the task by actively touching an object that entered the movement field of either the positive-valued "Go" whisker or the negative-valued "NoGo" whisker (Fig 1A, left). Mice reported the presence of the positive-valued stimulus by licking a port to receive a water reward (Hit response, Fig 1A, right). If mice licked the water port during the presentation of the negative-valued stimulus (False alarm), they were forced to locomote 2–4 times the normal distance to start the next trial. Mice took 10–20 days to reach consistent performance and performed similarly well on the day of recording (Fig 1B–1D). Task performance required active whisker touch (Fig 1E). Mice touched the positive and negative objects with similar frequency, and in nearly all trials (92%) responded with a latency greater than 0.3 s, which is greater than our window of analysis (Figs 1F, 1G and S1D). While mice performed the task,

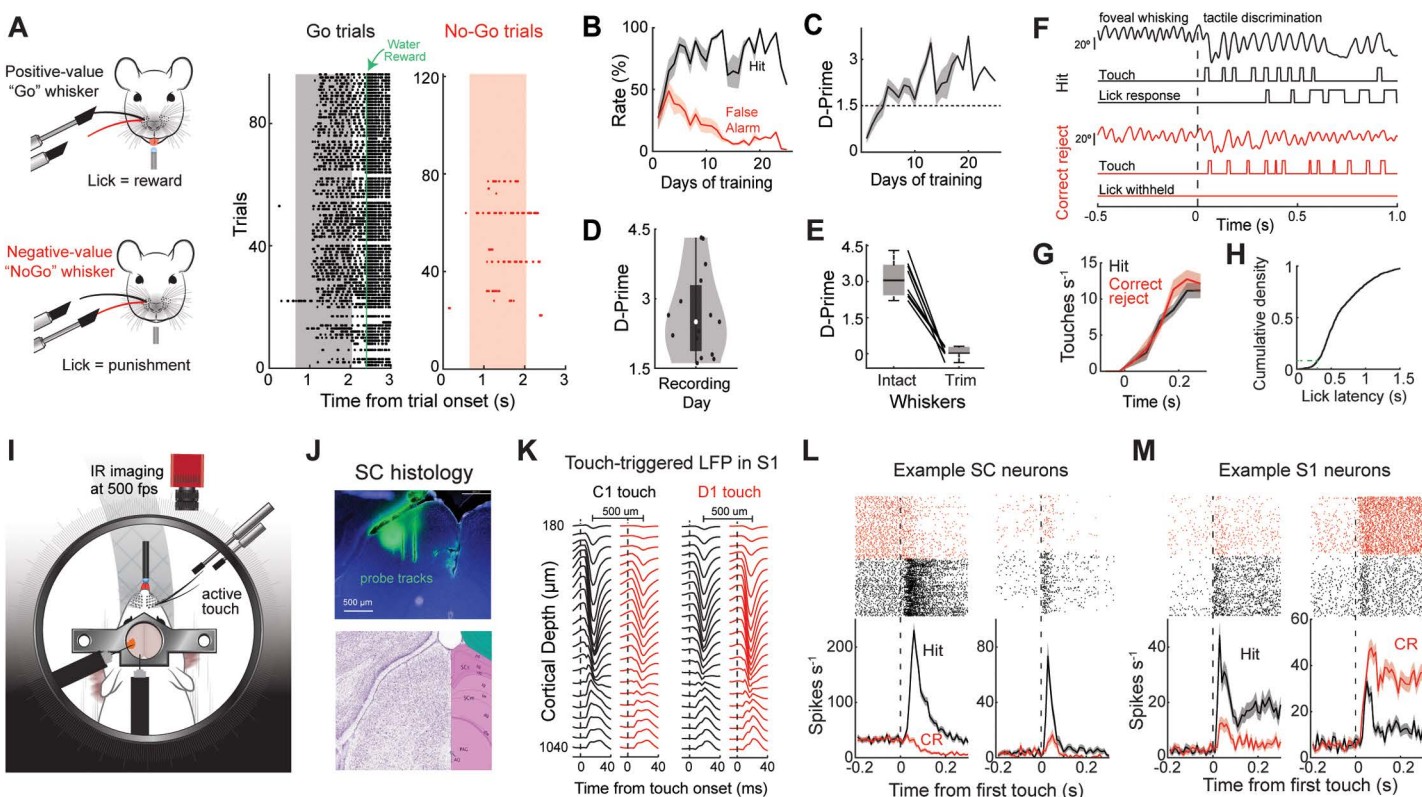

**Fig 1. Mapping value-based sensory associations in the somatosensory whisker system.** (**A**) Left, illustration of the positive- and negative-valued single whisker stimuli. A lick response to the positive stimulus was associated with reward (water) while a lick response to the negative stimulus was associated with punishment (extra locomotor delay). Right, rasters of lick timing during Go and No-Go trials. The touch window is shaded. (**B**) Mean Hit and False alarm rate as a function of days of training (16 mice). (**C**) Behavioral d-prime as a function days of training. (**D**) Behavioral d-prime on the day of the recording (11 mice). E) Behavioral d-prime on the day the whiskers were trimmed off (8 mice). (**F**) Whisker angle, touch duration, and lick responses during example Hit and Correct Rejection (CR) trials. (**G**) Mean touch rate during the Hit and CR trials across 15 behavioral sessions in 11 mice. (**H**) cumulative density of lick latencies (1655 trials across 15 sessions in 11 mice). (**I**) Illustration of the experimental system combining active tactile discrimination, whisker tracking, and electrophysiology in either the barrel cortex or superior colliculus. (**J**) Top, histological section of superior colliculus and three-shank silicon probe dye labeling. Bottom, Allen Brain Atlas of corresponding region of superior colliculus. (**K**) Touch-triggered local field potential (LFP) across two electrode shanks in somatosensory cortex (S1) during C1or D1 touch. Notice the change in LFP amplitude between the two stimuli, indicating a somatotopic map. (**L**) Rasters and histograms of firing rate in two SC neurons on trials when the animal responded to the positive stimulus (Hit, black) and withheld a response to the negative stimulus (CR, red). (**M**) Same as in G, except for two example S1 neurons. Error bars denote the standard error of the mean. See S1 Data (sheets 1D, 1E) for individual values of panels D, E.

we tracked whisker movement and recorded single-unit activity from populations of S1 or SC neurons using a high-density, three-shank silicon probe (Fig 1I). We targeted our electrodes to the intermediate and deep layers of SC whisker map using previously established stereotaxic coordinates and validated our placement by manually deflecting individual whiskers (Figs 1J and S1B) [17,25]. S1 recordings were made across all cortical layers and were guided to the whisker columns involved in the task using intrinsic signal imaging, with somatotopic locations confirmed post-hoc by examining touch-triggered local field potentials across the electrode shanks (Figs 1K, S1A, S1B). Aligning S1 and SC spiking to the onset of the first touch in each trial revealed short latency tactile responses with varying rates of adaptation and a preference for either the positive or negative whisker (Figs 1L, 1M, S1C). A total of 11 mice were trained to perform this version of the task. Most SC (76%, 549/727) and S1 (89%, 625/705) neurons displayed a significant tactile response to one or both whisker(s) ($p < 0.05$, one-way ANOVA with Tukey comparison).

## Task structure enhances positive stimulus bias in SC neurons

Both SC and S1 contain a somatotopic map of whisker space, where each whisker on the face activates a corresponding coordinate in each brain area [4,5,7]. We reasoned that if a brain area is faithful to this map, then the stimulus preference of its neural population would be symmetric for both whiskers. In line with this reasoning, we found that the stimulus preference of the S1 population was evenly distributed, reflecting only a small sampling bias (58%) towards the negative whisker (Correct Rejection (CR), Fig 2A). However, the stimulus preference of the recorded SC population was significantly skewed (70%) towards the positive whisker (Hit, Chi-squared test, $p < 0.001$). This skew for the positive whisker was similar across all SC recordings ($n$ = 8 mice; Chi-squared test, $p = 0.15$ for similarity among mice). Next, we sought to determine if the strength of stimulus bias depended on the stimulus preference of the neuron (Fig 2B, 2D). Among neurons that preferred the positive stimulus, the bias of the SC population was significantly greater than the bias of the S1 population (Fig 2C; $p = 4e^{-9}$, one-way ANOVA with Tukey comparison). However, among neurons that preferred the negative stimulus, bias among S1 and SC neurons was equivalent and indistinguishable from the positive preferring S1 population (Fig 2D, E, $p > 0.9$, one-way ANOVA with Tukey comparison). These effects were consistent across all mice and could not be

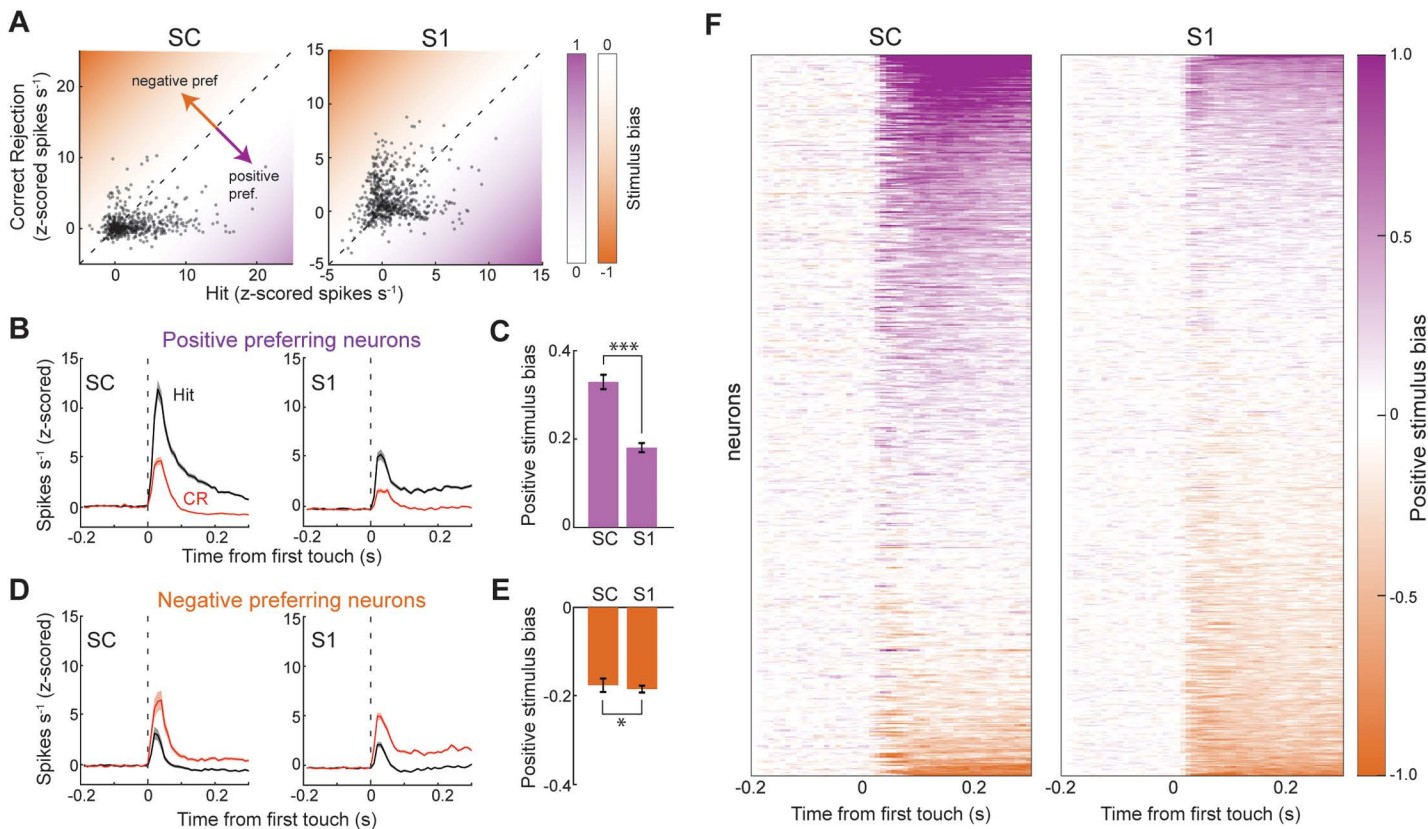

**Fig 2. SC neurons are strongly biased towards the positive stimulus.** (**A**) Scatter plot comparing neuronal firing rates (z-scored) during Hit and Correct Rejection (CR) trials (549 neurons in 8 mice in SC; 625 neurons from 7 mice in S1). (**B**) Histograms of firing rate during Hit and CR trials in neurons that preferred the positive stimulus (386 neurons in 8 mice in SC; 263 neurons from 7 mice in S1). (**C**) Mean positive stimulus bias in positive preferring SC and S1 neurons ($p = 7.7e^{-11}$, rank sum test). (**D**) Histograms of firing rate (z-scored) during Hit and CR trials in neurons that preferred the negative stimulus (163 neurons in 8 mice in SC; 362 neurons from 7 mice in S1). (**E**) Mean positive stimulus bias in negative preferring SC and S1 neurons ($p = 0.01$, rank sum test). (**F**) Heat maps of positive stimulus bias across the SC and S1 populations. Error bars denote the standard error of the mean. See S1 Data (sheets 2C, 2E) for individual values of panels C, E.

explained by differences in touch rate between conditions (S1E Fig). Therefore, SC neurons had a uniquely large positive stimulus bias, as evident in the population distribution (Fig 2F). This unique stimulus bias was evident within the first 50 ms of the response, indicating that feedforward sensory processing provided an important contribution to the effect (S1F Fig). Consistent with the sensory origin of the observed effects, SC activity in these neurons was unmodulated around lick initiation, and neuronal firing rates on false alarm trials were significantly smaller than Hits (S1G–S1I Fig).

Next, we hypothesized that stimulus bias in SC neurons was enhanced by the task structure of selecting one stimulus while ignoring the other (stimulus selection). To test this possibility, we trained a separate cohort of mice to detect and associate both whisker stimuli with reward, causing the two stimuli to have equal value and elicit equivalent behavioral responses (Figs 3A and S2). A total of 10 mice were trained for this task, of which 6 SC and 6 S1 recording sessions were performed. The majority of SC (67%, 332/449) and S1 (77%, 498/649) neurons displayed a significant tactile response to either or both stimuli ($p < 0.05$, one-way ANOVA with Tukey comparison). Interestingly, we now found that the stimulus preference of the SC population was more evenly distributed between both whiskers (Fig 3B; SC: 52% whisker 1 preferring; S1: 42% whisker 1 preferring). More importantly, stimulus bias of the SC population was now similar for both whiskers and slightly smaller than S1 (Fig 3C–3G, $p = 0.005$, rank sum test, bias for both whisker stimuli combined). This rules out sampling bias as the main driver for the large positive stimulus bias of SC neurons in Fig 1. Taken together, these data reveal that stimulus bias in SC, but not S1, was enhanced by the process of preferentially selecting one stimulus while ignoring the other.

## Spike suppression during correct rejection trials enhances positive stimulus bias in SC neurons

Next, we sought to identify the temporal dynamics underlying the large positive stimulus bias in SC neurons. First, we noticed that the CR response in SC neurons rapidly changed over time, with firing rates initially increasing, but then quickly decreasing, often to a level below the pre-stimulus baseline (Fig 4A). This delayed suppression of the CR response appeared to contribute to an increasing stimulus bias in SC; however, in S1, stimulus bias was constant over time (Fig 4B, 4C). To delineate the relationship between neural dynamics and stimulus bias, we quantified the bias of the Hit and CR responses relative to the pre-stimulus baseline. In the SC population, the CR response was initially positive but then decreased at a rate much faster than the Hit response (Fig 4D). The CR response became negative ~70 ms post first touch and reached its asymptote ~50 ms later, which coincided with the largest positive stimulus bias in the SC population (shown in Fig 4C). Conversely, in the S1 population, the Hit and CR responses increased and decreased at similar rates, leading to a constant stimulus bias over time (shown in Fig 4C). This indicates that spike facilitation during Hit trials combined with spike suppression during CR trials enhances positive stimulus bias in SC neurons. Stimulus bias over time and the underlying neural dynamics of negative preferring neurons was similar between SC and S1 (S3 Fig).

We reasoned that if spike suppression during CR trials enhances stimulus bias in SC, then neurons with a higher baseline firing rate could have a higher stimulus bias, due to their potentially greater dynamic range. In support of this logic, we found that baseline firing rate was positively correlated to positive stimulus bias in SC but not S1 neurons, particularly for the later period of the response, when spike suppression was greatest (Fig 4E). To determine the contribution of the Hit and CR responses to this trend, we plotted their touch-evoked firing rates during the late sensory period (70–300 ms from first touch, Fig 4F). In the SC

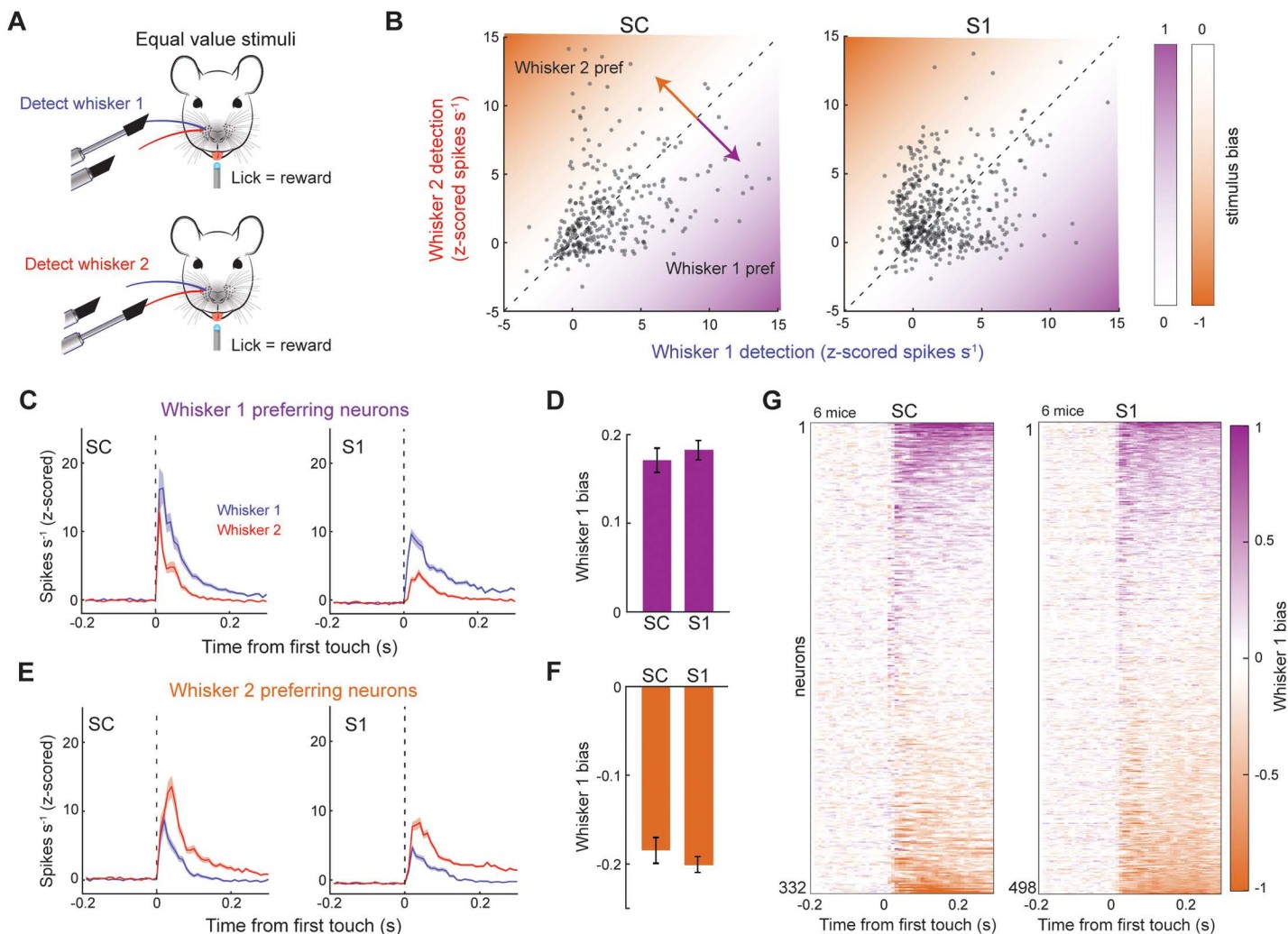

**Fig 3. Equal valued sensory associations reduce stimulus bias in SC.** (**A**) Illustration of the task structure. Licking in response to either of the single-whisker stimuli was paired with reward. (**B**) Scatter plots comparing neuronal firing rates (z-scored) during the detection of the two different single whisker stimuli (332 neurons in 6 mice in SC; 498 neurons from 6 mice in S1). (**C**) Histograms of firing rate (z-scored) in neurons that preferred the whisker 1 stimulus (172 neurons from 6 mice in SC; 210 neurons in 6 mice in S1). (**D**) Whisker 1 stimulus bias in neurons that preferred the whisker 1 stimulus. (**E**) Histograms of firing rate (z-scored) in neurons that preferred the whisker 2 stimulus (160 neurons in 6 mice in SC; 288 neurons in 6 mice in S1). (**F**) Whisker 1 stimulus bias in neurons that preferred the whisker 2 stimulus. (**G**) Heat maps of whisker 1 bias among the population of S1 and SC neurons. Error bars denote the standard error of the mean. See S1 Data (sheets 3D, 3F) for individual values of panels D, F.

population, neurons with higher baseline firing rates displayed larger increases in their Hit response, while also displaying larger decreases in their CR response. Therefore, the gap between the Hit and CR responses widened in neurons with larger baseline firing rates. Conversely, in the S1 population, the gap between the Hit and CR responses did not change as a function of baseline firing rate. Taken together, these data reveal that SC neurons with a higher baseline firing rate are more strongly biased towards stimuli with a positive association between action (licking) and outcome (water reward). Touch suppression and its interaction with baseline spike rate was significantly reduced in animals performing the stimulus detection task (S3F Fig), suggesting that the suppression of high firing rate neurons could be an important mechanism for ignoring unwanted stimuli.

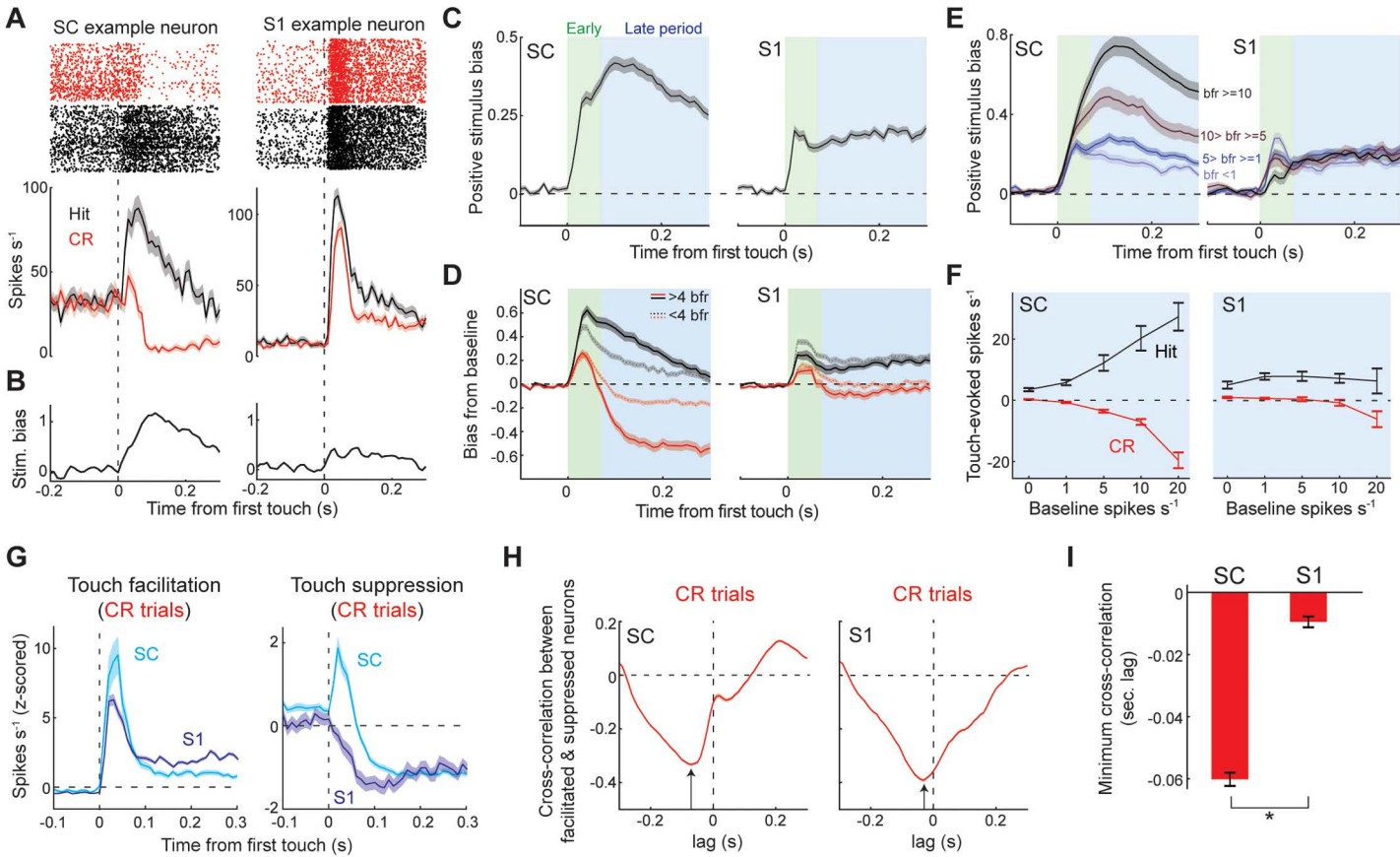

**Fig 4. Stimulus bias in SC is controlled by spike facilitation and spike suppression in single neurons.** (**A**) Rasters and histograms of mean firing rate in an example SC and S1 neuron during Hit and Correct Rejection (CR) trials. (**B**) Positive stimulus bias in the same example neurons. (**C**) Mean positive stimulus bias relative to first touch among the positive preferring SC and S1 neurons (386 neurons in 8 mice in SC; 263 neurons from 7 mice in S1). (**D**) Mean bias relative to baseline firing rate during Hit (black) and CR (red) trials in positive preferring SC (8 mice) and S1 (7 mice) neurons. Baseline firing rate ≥4, *n* = 186 neurons in SC and 117 in S1. Baseline firing rate < 4, *n* = 200 neurons in SC and 146 in S1. (**E**) Mean positive stimulus bias in positive preferring SC and S1 neurons where the populations are sorted by baseline firing rate (For SC: 107 neurons in 8 mice in bfr ≥ 10 group; 60 neurons in 8 mice in 10 > bfr ≥ 5 group; 131 neurons in 8 mice in 5 > bfr ≥ 1 group; 88 neurons in 8 mice in bfr < 1 group; For S1: 60 neurons in 7 mice in bfr ≥ 10 group; 42 neurons in 6 mice in 10 > bfr ≥ 5 group; 99 neurons in 7 mice in 5 > bfr ≥ 1 group; 62 neurons in 5 mice in bfr < 1 group). (**F**) Mean touch-evoked spike rates during the late period of the sensory response among populations grouped according to their baseline firing rate. (**G**) Population firing rates during CR trials in neurons that were significantly (*α* = 0.05, *t*-test) touch facilitated (left) or touch suppressed (right) by the negative stimulus (Facilitated: 110 SC neurons in 8 Mice, 283 S1 neurons in 7 mice; Suppressed: 133 SC neurons in 8 mice, 55 S1 neurons in 7 mice). (**H**) Cross-correlograms between the facilitated and suppressed firing rates during CR trials. (**I**) Bar graphs showing a significantly greater lag in SC neurons, as determined by the minimum cross-correlation, between touch facilitation and suppression (*p* < 0.001, one-way ANOVA with Tukey comparison). Error bars denote the standard error of the mean. See S1 Data (sheet 4I) for individual values of I.

Next, we investigated whether spike suppression in SC neurons was computed locally, or potentially through a separate pathway. To do so, we compared the timing of spike facilitation and suppression, reasoning they would occur simultaneously and with a short latency if computed locally. First, we plotted the firing rates of neurons that were significantly touch facilitated or suppressed by the negative stimulus, determined by their change in firing rate relative to baseline (Fig 4G). In the touch facilitated neurons, the onset of facilitation was immediate in both S1 and SC, suggesting that both brain areas get strong feed-forward excitation (Fig 4G, left). However, in the touch suppressed population, the onset of suppression was delayed in SC, suggesting that it could be computed from a different circuit (Fig 4G, right). To quantify the lag between facilitation and suppression, we cross-correlated the firing rates of the suppressed and facilitated neurons, on a mouse-by-mouse and

neuron-by-neuron basis (Fig 4H). In S1, firing rates were most anti-correlated with only a 10 ms lag, indicating that touch facilitation and suppression had nearly identical temporal dynamics. In SC, firing rates were most anti-correlated with a −60 ms lag, indicating that touch suppression occurred 60 ms after touch facilitation (Fig 4I).

## Task engagement and lick preparation enhances SC sensory processing

The SC is known to play a critical role in sensory-guided movements [11,12,26–28]. Given this knowledge, we hypothesized that tactile responses in SC neurons are influenced by whether the animal has the opportunity to behaviorally select the positive stimulus. To test this hypothesis, we compared sensory responses on Hit trials to when the animal Missed (failed to respond) or was experimentally denied the opportunity (water port physically displaced) to respond (Figs 5A, S4A–S4C). Miss trials were sporadic and primarily occurred in the latter half of the recording session, while Away (water port displaced) trials were experimentally controlled and occurred at regularly spaced intervals (Fig 5B). We focused our analysis on neurons that preferred the positive stimulus, given their strong stimulus bias and potential for influencing the decision to lick. In many SC neurons, the response to the positive stimulus was often smaller on Miss (48%, 133/276) trials and during the Away (38%, 102/268) condition, as shown in an example neuron and in population statistics (Fig 5C–5G). In S1, a change in sensory responsiveness on Miss trials was also apparent (55%, 75/136), while the Away condition had an infrequent (17%, 30/178) effect (Fig 5C–5G). Positive stimulus bias in SC neurons significantly decreased on Away trials, while in S1 the effect was significant but very small (Fig 5H). Despite this decrease during Away trials, the stimulus bias of positive preferring SC neurons was still uniquely large (Fig 5I). To determine if the early sensory response decreased during Away trials, we restricted our analysis to the first 50 ms post-touch. During this early period, the Hit response and positive stimulus bias significantly decreased ($p = 2.8e^{-6}$ for Hit response; $p = 1.7e^{-7}$ for stim bias; $n = 237$, signed rank test). Differences in touch strength or locomotion speed could not explain these effects (S4D, S4E Fig). SC neurons that were modulated by Miss were also equally modulated by Away, indicating that task disengagement and water port removal reduced the tactile sensitivity of the same population of SC neurons (Fig 5J). Sensory processing in negative preferring neurons was largely unaffected by behavioral context (S4F–S4K Fig).

## Baseline SC firing rates predict task performance and reaction times

Evidence suggests that perceptual decisions are made when neuronal firing rates reach a threshold [29]. Therefore, one potential mechanism for speeding up decision-making is by decreasing the distance of choice-related neurons from their threshold [30]. To examine if "distance to threshold" is modulated by behavioral context, we examined baseline firing rates during Engaged (Hit and CR), Miss, and Away trials (Fig 6A). On Miss trials, when mice failed to select the upcoming positive stimulus, pre-stimulus baseline firing rates in both SC and S1 neurons were lower (Fig 6B). Interestingly, during Away trials, when the opportunity to select the stimulus was experimentally removed, only SC showed an appreciable decrease in baseline firing rate (Fig 6C). A change in locomotion speed could not explain these effects (S5A, S5B Fig). At the single neuron level, SC neurons were more than twice as likely as S1 neurons to display a significant reduction in baseline activity during the Away condition (Fig 6D). Therefore, baseline SC activity was more strongly related to the opportunity to respond (Home versus Away), while S1 activity may be a stronger reflection of task disengagement associated with Miss trials. The baseline firing rate of negative preferring SC neurons were largely unaffected by the Away condition, indicating that positive preferring neurons may represent a specialized population of movement-related neurons (S5D–F Figs).

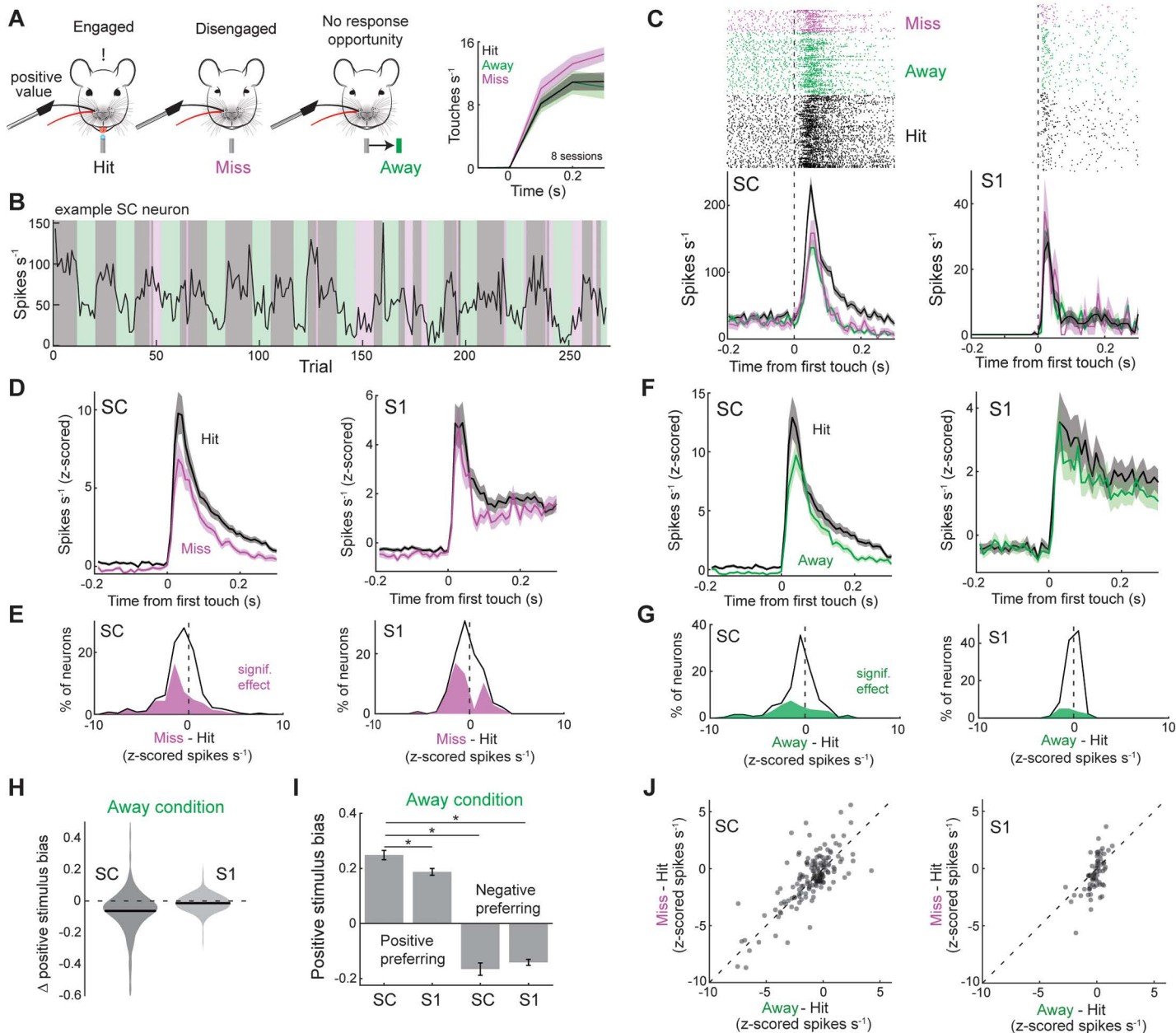

**Fig 5. Task engagement increases positive stimulus bias in SC neurons.** (**A**) Diagram illustrating the three different behavioral contexts during positive whisker stimulation. Right, touch rate during the different contexts. (**B**) Firing rates, calculated 0–300 ms post first touch, of an example neuron during the different behavioral contexts. (**C**) Rasters and histograms of firing rate in an example SC and S1 neuron across the different behavioral contexts. (**D**) Mean firing rate in SC and S1 neurons significantly modulated by the Miss condition (133/276 neurons in 6 mice in SC; 75/136 neurons from 4 mice in S1; $p < 0.05$, Mann–Whitney U-test). (**E**) Population histogram of change in firing rate (z-scored) during the Miss condition for all recorded SC and S1 neurons (Population test: $p = 2.6e^{-8}$ in SC & $p = 7.4e-$ in S1, one sample $t$-test). Neurons that displayed a significant change are denoted in magenta. (**F**) Mean firing rate in SC and S1 neurons significantly modulated by the Away condition (102/268 neurons in 5 mice in SC; 30/178 neurons from 3 mice in S1; $\alpha < 0.05$, Mann–Whitney U-test). (**G**) Population histogram of change in firing rate during the Away condition for all recorded SC and S1 neurons ($p = 6.5e^{-7}$ in SC & $p = 1.5e^{-2}$ in S1, one sample $t$-test). Neurons that displayed a significant change are denoted in green. (**H**) Change in stimulus bias in positive preferring SC and S1 neurons during the Away condition ($p = 9.5e^{-12}$, 268 neurons in 5 mice in SC; $p = 9.4e^{-4}$, 178 neurons from 3 mice in S1, two-sided $t$-test). (**I**) Positive stimulus bias calculated during Away trials. Positive preferring SC neurons had significantly greater absolute bias than all other groups (one-way ANOVA with Tukey comparison, $p < 0.05$). (**J**) Scatter plot comparing the effect of the Away and the Miss conditions in neurons that experienced all three behavioral contexts (158 neurons in 3 mice in SC; 67 neurons from 1 mouse in S1). See S1 Data (sheets 5H, 5I) for individual values of H, I.

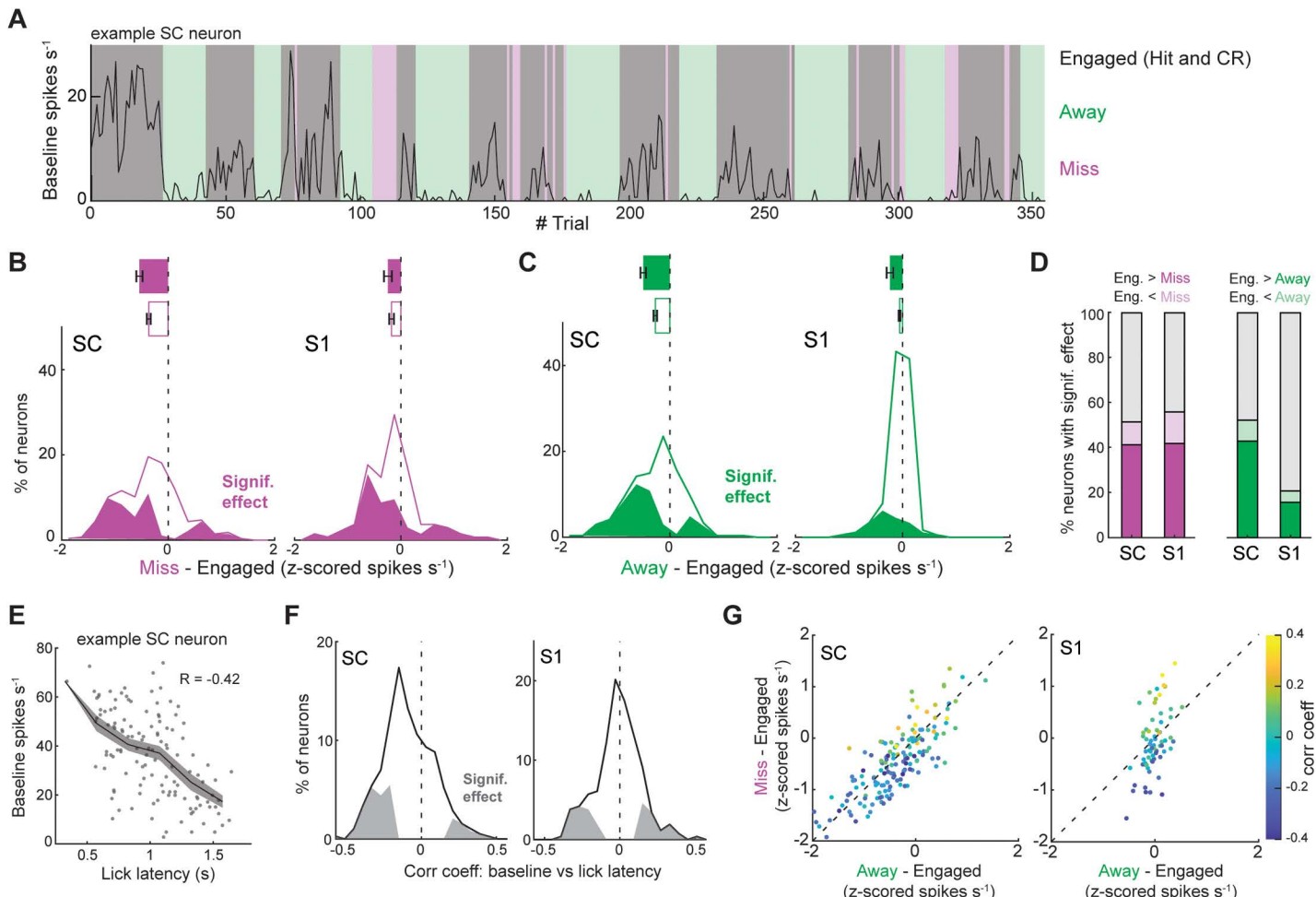

**Fig 6. Spontaneous firing rates predict task performance and response latency.** (**A**) Baseline firing rate of an example neuron that during an engaged, away, or miss trial. (**B**) Population histogram of change in baseline firing rate during the Miss condition ($p = 1.0e^{-20}$ in SC; $p = 1.2e^{-4}$ in S1, one sample $t$-test). Neurons with a significant effect are denoted in the solid magenta distribution (142/276 neurons in 6 mice in SC; 76/136 neurons from 4 mice in S1; $p < 0.05$, Mann–Whitney U-test. (**C**) Population histogram of change in baseline firing rate during the Away condition ($p = 6.90e^{-15}$ in SC; $p = 7.95e^{-4}$ in S1, one-sample $t$-test). Neurons with a significant effect are denoted by the solid green distribution (140/268 neurons in 5 mice in SC; 37/178 neurons from 3 mice in S1; $\alpha < 0.05$, Mann–Whitney U-test). (**D**) Percentage of neurons with a baseline firing rate significantly modulated by the Miss or Away condition ($p < 0.05$, Mann–Whitney U-test). (**E**) Spearman correlation between baseline firing rate and lick latency in an example SC neuron. (**F**) Histogram of the correlation between baseline firing rate and lick latency in the population of SC and S1 neurons (95/386 neurons in 8 mice in SC; 68/263 neurons from 7 mice in S1; $\alpha < 0.05$, Pearson's correlation; Population test: $p = 1.4e^{-21}$ in SC & $p = 0.29$ in S1, one sample $t$-test). Neurons with a significant correlation are denoted by the gray distribution. (**G**) Scatter plot comparing the effect of Miss and Away conditions on baseline firing rate. The color axis denotes the correlation of each neuron's baseline firing rate with lick latency (158 neurons in 3 mice in SC; 67 neurons from 1 mouse in S1). Error bars denote the standard error of the mean.

Next, we reasoned that if SC neurons are involved in driving perceptual decisions (i.e., to lick), then their baseline firing rates should correlate to reaction time (Fig 6E). In line with this reasoning, we saw that higher baseline spike rates in SC but not S1 neurons were associated with shorter response latencies (Fig 6F). Interestingly, SC neurons with baseline firing rates that were modulated by behavioral context (Miss/Away) were also correlated to reaction time (Fig 6G).

## Discussion

In this study, we show that primary S1 contains a faithful map of whisker space, where stimulus bias is predominantly controlled by a stable representation of somatotopic space that

persists across behavioral contexts. Conversely, we show that SC contains a value-based map of whisker space that is strongly biased towards positive stimuli and enhanced by stimulus selection. Moreover, the spontaneous firing rate of positive preferring SC, but not S1, neurons predicted the reaction time of licking. Taken together, these data argue that SC neurons play a key role in transforming somatotopic features into value-based actions. What circuits could be critical to this process?

Both S1 and SC neurons receive input from brain areas involved in sensation, action, and associative learning. In the somatosensory whisker system, the SC gets monosynaptic excitation from trigeminal neurons in the brainstem, while S1 receives brainstem excitation via primary and secondary thalamic nuclei [13,17,31–34]. Hypothetically, SC-projecting trigeminal neurons could have greater stimulus bias than their thalamic-projecting neighbors, thereby providing the substrate for greater stimulus bias in SC. However, whisker-sensitive neurons in the brainstem (SpV) that target the secondary thalamus (Po) also send axons collaterals to SC, indicating a shared ascending sensory representation [35]. If the shared pathway to SC and Po is agnostic to stimulus value, our data suggests that local processing in SC is sufficient for value-modulated somatosensory processing, due to the large, significant bias of the initial (within 50 ms post-touch) tactile response (Figs 2B and S1F). If the ascending pathway is biased by stimulus value, then a subpopulation of Po→S1 neurons should mirror our results in the SC. This possibility is intriguing, given the known SC→Po pathway that augments tactile responses in S1 [18], and potentially forms a layer-specific value-modulated loop between cortex and midbrain [33,36]. Nonetheless, the contribution of this ascending secondary pathway to sensory maps in S1 remains unclear, yet its activity is known to be modulated by decision-making [37,38]. A technique that enables a cell-type and projection-specific (Po→S1) analysis of S1 neurons is needed to fully resolve this question. Anatomically, the primary ascending pathway from brainstem to S1 is largely distinct from the secondary pathway that involves the SC, yet the primary and secondary brainstem nuclei are known to target both thalamic nuclei, albeit with a clear bias toward their respective targets [1,35]. Ultimately, it remains unclear if the anatomical and functional segregations in the ascending pathways are sufficient to generate the functional differences observed in our study. Future investigations into somatosensory processing in the brainstem are needed to clarify the origin of value-modulated processing in SC and other brain areas.

We discovered that spike suppression on CR trials drove the large positive stimulus bias in SC neurons. Spike suppression emerged during the later period (>70 ms) of the sensory response and was preceded by a short window of spike facilitation, suggesting the presence of rapid sensory excitation followed by strong inhibition. SC neurons have large excitatory receptive fields, often facilitated by several different whiskers [39,40]. We hypothesize that the substantia nigra pars reticulata (SNr), a prominent inhibitory input to the SC [41–44], counterbalances this broad sensory excitation to prevent unwanted stimuli from driving actions. This hypothesis is supported by the causal role of the SNr→SC circuit in controlling tongue-mediated target selection [42,45]. Furthermore, the delayed onset of SC suppression in our task mirrors the timing of SNr activation during low-value stimulus presentation [46], and reflects the serial delays imposed by the polysynaptic cortico-striatal pathway [47–49]. Nonetheless, a local circuit mechanism is also possible, whereby recurrent inhibition overcomes feedforward excitation. Future research focused on dissecting the contribution of the cortico-striatal pathway will provide critical insight into the mechanisms of value-based stimulus bias in SC [42,50–53].

Behavioral context had a significant effect on positive stimulus bias in SC neurons. Stimulus bias significantly decreased when mice were trained to perform detection, where both whisker stimuli elicited the same lick response in association with the same reward (Fig 2).

Therefore, the task structure of selecting one stimulus while ignoring the other amplified bias in SC, but not S1. Sensory suppression to non-preferred stimuli was significantly weaker during stimulus detection, suggesting that SC suppression could be important for ignoring unwanted stimuli. Given this insight, we hypothesized that the animal's readiness to perform stimulus selection shapes sensory processing in SC neurons. To test this hypothesis, we physically displaced the water port, removing the opportunity for mice to respond to the positive stimulus. During this period, mice whisked against the object normally but never licked or extended their tongue, indicating they were aware of the experimental constraint. Interestingly, the Hit response in positive preferring SC neurons decreased, leading to a significant decrease in SC stimulus bias (Fig 5). Therefore, lick preparation increased tactile sensitivity in SC neurons. Similarly, preparation for eye movements has been shown to increase the visual sensitivity of monkey SC neurons and enhance task performance in humans [21,54], presumably by directing attention towards the location of the planned movement. In our study, the presence of the water port, which was in close contact with the lips and surrounding hairs, presumably prepared the animal to lick and directed spatial attention towards the positive whisker. The initial (50 ms) SC sensory response was altered by this preparation, indicating that ongoing internal dynamics shape feedforward processing in SC neurons. In the visual system, similar effects have been observed not only in SC, but in higher-order visual and parietal cortex [55–57]. It is important to note that lick preparation also modified the later period (100–300 ms) of the tactile response, which was unlikely caused by lick initiation, since mice licked with a much longer latency than our analysis window, and none of our recorded neurons increased their activity around the onset of lick.

A neural correlate of task engagement was also evident in the pre-stimulus firing rate of SC neurons. When the water port was periodically removed, we observed a significant decrease in baseline spiking exclusively in the positive preferring SC population. With the water port in place, the pre-stimulus firing rate of these neurons was significantly correlated to the animal's reaction time, suggesting that these neurons are important for transforming tactile stimuli into actions. A similar class of SC neurons has been observed in monkeys, where the pre-stimulus firing rate of "build-up" neurons predicted saccadic reaction time [58–60]. This modulation in pre-stimulus firing rates is consistent with the SC mediating a decision threshold [61–64]. At the population level, S1 neurons were weakly affected by lick preparation. However, during Miss trials, when the animal was presumably disengaged from the task, we also observed a decrease in S1 baseline firing rates. Therefore, task disengagement during Miss trials may coincide with global changes in brain state that spread throughout primary sensory cortex [65–68]. The influence of movement preparation may predominantly occur in downstream areas more closely associated to action [69].

In this study, we reveal that the representations of sensory space in S1 and SC are differentially modulated by behavioral context. We show that active engagement in value-based stimulus selection greatly biases sensory maps in SC, but not S1, towards the positive stimulus. The circuit mechanisms supporting these differences are unknown, likely involving bottom-up or local changes in feedforward processing as well as top-down influences from cortex and basal ganglia. Understanding the contribution of these circuits to value-based sensory processing will provide critical insight into the neural mechanisms underlying intelligent behavior.

## Methods

### Experimental model

This study utilized a total of 23 mice of both sexes with CD1 background. The mice were housed socially in groups of five or fewer per cage. They were kept in a controlled

environment with temperatures ranging from 68 to 79 °F and humidity levels between 40% and 60%. The mice were maintained on a reverse light-dark cycle (12:12 h), with experiments conducted during their subjective night. All surgical and experimental procedures were approved by the Purdue Institutional Animal Care and Use Committee (IACUC, 1801001676A004) and the Laboratory Animal Program (LAP).

## Head-plate surgery

Each mouse was fitted with a custom-designed aluminum headplate for head fixation. The headplates included a circular opening with a 9.4 mm diameter to allow access to both S1 and SC regions. During the procedure, the animals were anesthetized using 3%–5% isoflurane gas. Their body temperature was maintained with a heating pad, and their respiratory rate was continuously monitored to ensure stable anesthesia. To prevent eye dryness during surgery, ointment was applied. The skin and fur overlying the skull were disinfected with 70% ethanol and betadine to minimize the risk of infection. Lidocaine was injected under the scalp and using sterilized surgical instruments, an incision was made along the midline of the scalp. Next, the skin and underlying tissue were carefully removed to expose the skull. The headplate was positioned over the skull with access points for S1, and SC regions through the opening as measured from the Bregma/Lamba coordinates. To secure the headplate, *Liquivet* tissue adhesive and *Metabond* dental cement were applied to the skull and wound edges. Once the headplate was firmly attached and the dental cement had cured, post-operative analgesic was administered, and the mouse was monitored during recovery.

## Behavioral training

Mice were trained in one of two tasks: (1) a whisker discrimination task and (2) a whisker detection task. In the whisker discrimination task, one whisker was associated with a reward and the other whisker was associated with a punishment. In contrast, in the whisker detection task, both whiskers were associated with a reward of equal value.

Two days after headplate implantation, mice were habituated to running on circular treadmills. This habituation phase consisted of one session per day, lasting 1–2 h, for 4–7 days. This allowed the mice to become accustomed to locomoting on the circular treadmill. Following habituation, mice were placed on water restriction to increase their motivation for the water reward. All whiskers, except an adjacent whisker pair (B1 & C1 or C1 & D1), were reduced in length to exclude their use during the task.

The first stage of training involved classical conditioning. Each trial began with the mouse running two rotations on the circular treadmill, which was followed by the onset of the stimulus. The stimulus consisted of the protrusion of one of two pneumatically controlled touch surfaces (SMC pneumatics), designed to interact with one of the two targeted whiskers (B1/C1 or C1/D1) when the mouse was actively sampling the stimulus space. The stimulus remained in the whisker field for 1.5 s, after which water reward was delivered for the rewarded whisker. After the mice demonstrated anticipatory licking, they progressed to the second stage of training: operant conditioning. In this stage, the water was delivered only if the mouse licked the water port during presentation of the rewarded stimulus. If the mouse licked during presentation of the unrewarded whisker, it was punished by being required to run 2–4 times the normal distance to start the next trial.

## Intrinsic signal imaging for localizing whisker barrels in S1

Two days prior to S1 electrophysiology recording, intrinsic signal optical imaging was performed to locate the barrel columns contralateral to the trained whiskers. During the entire

imaging session, mice were anesthetized using 1% isoflurane and xylazine (0.3% mg/kg). The skull over S1 was thinned using a dental drill until the superficial vasculature became visible when moistened with artificial saline. 100% silicone oil was then applied to the head well and a cover slip was placed over it to create a flat imaging plane. Under green LED illumination an image of the superficial vasculature was obtained using a CCD camera (Retiga R1, *QIM-AGING*). Under red LED illumination, changes in fluorescence during whisker stimulus were calculated relative to a baseline period. During the stimulus period, the C1 and D1 whiskers were wiggled using piezoelectric actuators at 20 Hz for 4 s. Intrinsic signals acquired at a frame rate of 10 Hz over 15 trials with a 10-s intertrial interval were averaged. The sites with greatest change in fluorescence were marked and registered with the superficial vasculature to guide electrode insertion (S1E fig). The skull surface was cleaned, covered with a soft silicone gel (Dowsil) and then a hard-setting silicone (Kwik-Cast, World Precision Instruments). Mice were monitored during recovery from anesthesia and provided analgesics subcutaneously (Carprofen 5 mg/kg).

## In vivo electrophysiology

One day prior to the recording session, mice were briefly anesthetized with isoflurane to perform craniotomies over S1 and SC. For S1, a rectangular window of skull (~1 mm × ~0.5 mm) was carefully removed using a scalpel to expose the brain area overlaying the two whisker barrels columns identified by intrinsic imaging. For SC, a 1 mm circular craniotomy was performed 4 mm posterior and 1.5 mm lateral from bregma. Throughout the procedures, the exposed brain areas were kept moist with ACSF and covered with *Dowsil* silicone gel and *Kwik-Cast*. On the first day of the recording session, a 128-channel, 3-shank *Neuronexus* probe was targeted into the S1 barrels identified by intrinsic imaging. The probe was lowered into the cortex at a rate of 75 μm/min using a *NewScale* micromanipulator. Neural data and experimental signals were acquired at a 20 kHz sampling rate. On the second day of the recording session, the *Neuronexus* probe was targeted to the whisker receptive region of the intermediate and deeper layers of SC. The receptive field of recorded neurons was mapped by deflecting individual whiskers to identify which whiskers elicited the greatest response. If the electrode missed the target (C1/D1/B1), it was repositioned based on somatotopic coordinates.

## Spike sorting

Spike sorting was carried out using the Kilosort2 package in *MATLAB*, followed by manual curation with the Phy2 GUI (https://github.com/cortex-lab/phy) [70]. To classify spike clusters as single units, we evaluated their spike waveforms, electrode locations, and cross-correlograms to merge, split, or delete clusters as necessary. The spike clusters were validated as signal units based on waveform shapes and auto-correlograms. Only single units were included in all subsequent analyses presented in this paper.

## Trial structure and categorization

The training sessions were conducted in custom-built, sound-attenuated chambers with white background noise and no light. Each session lasted approximately 90 min. Trials began after the mouse had locomoted for two rotations on a circular treadmill. Following this, a touch surface protruded into the whisking field of one of the two whiskers, serving as the stimulus. The stimulus was presented for 1.5 s. If the mouse licked during this window in response to the rewarded stimulus, a water reward was delivered at the end of the response window. These trials were considered as Hit trials. If the mouse did not lick, it was considered a Miss trial. For a non-rewarded whisker stimulus, if the mouse withheld licking, the trial is considered

a CR trial or a false alarm (FA) trial otherwise. To account for satiation and disengagement, non-licking trials which occurred during a consecutive ($n \geq 5$) miss trials window are removed from the CR trials.

The Away condition was experimentally controlled via a stepper motor that moved the water port outside the reach of the mouse. All experiments started with the port in the Home position. After seven positive stimulus trials accumulated in the Home position, the water port was then moved into the Away position. After seven positive stimulus trials occurred in the Away position, the port was returned Home, and the process was repeated. The recording days were the only time the mouse ever experienced the Away condition.

### Whisker tracking & kinematics

The two intact whiskers were imaged at 500 fps for the entire duration of the trial using a high-speed IR camera (*Photonfocus DR1*). DeepLabCut (DLC) was utilized to label the whiskers in each frame, with 4 labels placed on each whisker. To train the DLC network, 200 frames spanning various stimulus conditions for each mouse were manually labeled and used for training the network with over 200,000 iterations. Whisker angles were calculated for each label with reference to a user-defined point on the face relative to the frame's vertical axis. Whisker curvature was calculated using the Menger curvature, applied to three points on the whisker at a time, with the mean of all possible combinations taken.

### Identifying touch times

The first onset of touch in each trial was identified as the time the piston entered the whisking field (S1F Fig). To identify individual touch times, a region of interest (ROI) was created at the user-defined interface of the whisker with the object surface. The touch time for each whisk cycle was determined by identifying when the whisker entered the surface ROI. The rare trials where the piston extended behind the whisking field and contacted during the retraction phase of the whisk cycle were excluded from analysis.

### Statistical analyses

**Stimulus bias.** All statistical tests were conducted in Matlab 2022 or later (Mathworks). To measure the selectivity of individual neurons to each stimulus, we calculated the stimulus bias at each time point using neuromeric d-prime. It is calculated as the difference in firing rates between two stimuli divided by the root mean square of standard deviation across these stimuli.

$$stimulus\ bias\ =\ \frac{\mu_A - \mu_B}{\sqrt{\frac{\sigma^2_A + \sigma^2_B}{2}}}$$

$\mu_A$ and $\mu_B$ are the mean spike rates in response to condition A and B respectively. $\sigma^2_A$ and $\sigma^2_B$ are the variances of spike rates across trials for condition *A* and *B*, respectively. This calculation was performed every 10 ms, providing a continuous estimation of bias throughout the response window.

**Hit and CR preferring neurons.** Touch responsive neurons were identified by testing for significant firing rate differences between the baseline period (1.5 s preceding touch onset) and the stimulus period (300 ms post-touch) using one-way ANOVA with Tukey post-hoc for multiple comparisons. To further categorize these neurons based on their stimulus preference, we calculated the mean stimulus bias within the 300 ms window following touch

onset. Neurons were then determined as either Hit-preferring or CR-preferring based on the direction of this value (positive/negative).

**Feature occupancy control.** To control trial-to-trial behavioral variance (run speed and touch strength variance), observations are sub-sampled such that any comparison made across stimulus conditions are from distributions with equal feature distributions. Results were created from the average of 20 resampling permutations.

**Cross-correlograms.** To find the time lag between stimulus driven facilitation and suppression, neurons were first tested for significant facilitation or suppression (0–300 ms post-touch) from baseline (1.5 s preceding touch onset) using a $t$-test. In each mouse, the trial-averaged z-scored touch response of every facilitated neuron was cross correlated with every suppressed neuron in that mouse (Matlab, xcorr). The lag of the minimum cross-correlation was used to determine temporal offset between touch facilitation and suppression, where a negative lag corresponds to touch suppression occurring after touch facilitation.

## Supporting information

**S1 Fig. Mice learn value based sensory associations using active whisker touch.** (**A**) Example images from intrinsic imaging of S1 to locate barrel columns corresponding to whiskers used in the task (left and middle). (**B**) Depth of recording neurons that significantly responded to a tactile stimulus .(**C**) D-Prime of expert mice pre and post whisker trim ($n = 8$ mice). The locations of highest intrinsic signal changes overlayed with image of the superficial vasculature to guide electrode placement (right). (**C**) Image analysis showing how the onset of first touch was determined from the time the object entered the active whisking field. (**D**) Mean touch rate for first 300ms of sensory period during hit and correct reject stimuli ($n = 15$ mice). (**E**) Mean positive stimulus bias of SC and S1 neurons relative to touch rate bias. Each point is from a single mouse. Touch bias was calculated the same as stimulus bias, except using touch rates instead of firing rates. (**F**) Positive stimulus bias using only the first 50 ms of the sensory response. SC positive stimulus bias was greater than all other biases (one-way ANOVA, Tukey comparison, $p < 0.001$). (**G**) Left, trial-averaged stimulus bias aligned to the onset of touch. Right, trial-averaged stimulus bias aligned to the onset of licking (268 neurons). (**H**) Top, trial-averaged firing rate in an example mouse during Hit, CR, and false alarm (FA) trials. Bottom, trial-averaged lick probability for the corresponding Hit and FA trials. (**I**) Mean firing rate of SC positive-preferring neurons during Hit, CR, and FA trials. CR and FA firing rates were statistically equivalent ($p = 0.3$, one-way ANOVA with Tukey comparison, 386 neurons in 8 mice for CR, 335 neurons in 7 mice for FA). See S1 Data (sheets S1F, S1I) for individual values of F, I.
(EPS)

**S2 Fig. Mice learn equal valued sensory associations.** (**A**) Example lick rasters from an expert mouse trained to lick in response to either of the two single whisker stimuli. (**B**) Average hit rates for both single whisker stimuli over training sessions ($n = 7$ mice). (**C**) Hit Rates during S1 and SC electrophysiology recordings ($n = 6$ mice). (**D**) Hit rates during both single whisker stimuli pre and post whisker trimming ($n = 4$ mice). (**E**) Mean touch rate for first 300 ms of sensory period during positive whisker 1 and 2 stimuli ($n = 10$ mice). (**F**) Percent of neurons in SC and S1 that preferred Whisker 1 according to shank location. See S1 Data (sheets S2C, S2D, S2E, and S2F) for individual values of C, D, E and F.
(EPS)

**S3 Fig. Stimulus bias of negative preferring neurons is similar between SC and S1.** (**A**) Negative stimulus bias in neurons that preferred the negative valued stimulus. (**B**) Same as in A except with the data grouped by baseline firing rate. (**C**) Bias from baseline in negative preferring neurons. Neurons were divided into two group according to their baseline firing

rate (bfr). (**D**) Mean touch-evoked firing rates during the late period of Hit and CR response in negative preferring neurons. (**E**) Mean touch-evoked firing rates during the early period of the Hit and CR response in positive preferring neurons. (**F**) Mean touch-evoked firing rates of positive/whisker 1 preferring neurons during stimulus selection (Hit vs. CR) and detection (Both Go). SC Touch-evoked firing rates during W2 detection trials were significantly different than firing rates during CR trials ($p = 2.5e^{-14}$, two-way ANOVA). The interaction between SC baseline spike rate and touch-evoked firing rate was significant among CR and W2 neurons ($p = 2e^{-9}$, two-way ANOVA). Sample sizes are the same as in Fig 4. Error bars denote the mean ± SEM. See S1 Data (sheets S3D, S3E, S3F) for individual values of D, E, and F.
(EPS)

**S4 Fig. Unexpected licking, locomotion, or touch quality cannot explain the effects of the Miss or Away conditions.** (**A**) Image frames of facial movement analysis for determining if the animal licked when the water port was Away. (**B**) Quantification of image pixel values for extracting lick rate during present and Away conditions. (**C**) Lick and grooming raster for the Present and Away conditions. (**D**) Change in spike rate in positive preferring neurons for all trials and for trials where the locomotion speed and curvature were subsampled to match the occupancy of the Miss condition. (**E**) Same as D, except for the Away condition. (**F**) Firing rates of negative preferring neurons during Hit and Miss trials. (**G**) Histogram of change in neuronal spike rates during the Miss condition. (**H**) Same as F, except for the Away condition. (**I**) Same as G, except for the Away condition. (**J**) Scatter plot of mean ± SEM touch rates calculated during Hit, Miss, and Away trials of each mouse in the study. (**K**) Change in positive stimulus bias in negative preferring neurons during the Away condition. See S1 Data (sheets S4B, S4J, S4K) for individual values of B, J, and K.
(EPS)

**S5 Fig. Baseline firing rate: locomotion speed control and the effect of behavioral context on negative preferring neurons.** (**A**) Change in baseline firing rate in positive preferring neurons, comparing the Miss effect calculated from all trials to the locomotion occupancy-controlled Miss effect. (**B**) Same as in A, except for the Away condition. (**C**) Histogram of the change in baseline firing rate in negative preferring neurons during the Miss condition. (**D**) Same as in C, except for the Away condition. (**E**) Bar graph of the percent of neurons that significantly increase, decrease, or do not change during the Miss or Away condition. (**F**) Correlation coefficient between baseline spike rate and lick latency in negative preferring neurons. (**G**) 3-dimensional scatter plot comparing the effect of Miss and Away on baseline spike rate and the corr. coeff. with latency in negative preferring neurons.
(EPS)

**S1 Data. An excel spreadsheet containing the individual data values for each of the figure panels that contain summary data.** The name of each sheet in the Excel file corresponds to a figure panel in the manuscript.
(XLSX)

## Acknowledgments

The authors would like to thank members of the Pluta lab and Edward Zagha for valuable input on the manuscript.

## Author contributions

**Conceptualization:** Scott R. Pluta.

**Data curation:** Suma Chinta, Hayagreev V. S. Keri.

**Formal analysis:** Yun Wen Chu, Suma Chinta.

**Funding acquisition:** Scott R. Pluta.

**Investigation:** Suma Chinta.

**Methodology:** Suma Chinta, Hayagreev V. S. Keri, Shreya Beri, Scott R. Pluta.

**Project administration:** Scott R. Pluta.

**Supervision:** Suma Chinta, Scott R. Pluta.

**Validation:** Hayagreev V. S. Keri.

**Visualization:** Yun Wen Chu, Suma Chinta, Hayagreev V. S. Keri, Scott R. Pluta.

**Writing – original draft:** Suma Chinta, Scott R. Pluta.

**Writing – review & editing:** Yun Wen Chu, Suma Chinta, Hayagreev V. S. Keri, Scott R. Pluta.

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
