## [Editor Report · Decision Letter 0]

27 Aug 2024

Dear Scott, 

Thank you for submitting your manuscript entitled "Stimulus selection drives value-modulated somatosensory processing in superior colliculus" for consideration as a Research Article by PLOS Biology. Apologies for the delay in getting back to you. I was waiting for advice from one of our editorial board members.

Your manuscript has now been evaluated by the PLOS Biology editorial staff and I am writing to let you know that we would like to send your submission out for external peer review. Unfortunately, we have not been able to receive advice from one of our academic editors on your study and have, therefore, not yet made a firm decision on whether the conceptual advance is sufficient for PLOS Biology. We will discuss this after review with one of our editorial board members and will be looking for strong reviewer support.

Once your full submission is complete, your paper will undergo a series of checks in preparation for peer review. After your manuscript has passed the checks it will be sent out for review. To provide the metadata for your submission, please Login to Editorial Manager (https://www.editorialmanager.com/pbiology) within two working days, i.e. by Aug 29 2024 11:59PM.

Kind regards,

Christian

Christian Schnell, PhD

Senior Editor

PLOS Biology

cschnell@plos.org

---

## [Decision Letter · Decision Letter 1]

10 Oct 2024

Dear Scott,

Thank you for your patience while your manuscript "Stimulus selection drives value-modulated somatosensory processing in superior colliculus" was peer-reviewed at PLOS Biology. It has now been evaluated by the PLOS Biology editors, an Academic Editor with relevant expertise, and by several independent reviewers. 

In light of the reviews, which you will find at the end of this email, we would like to invite you to revise the work to thoroughly address the reviewers' reports.

As you will see below, the reviewers think that the study is very well executed and provides important insights. While Reviewer 1 and 2 are most positive and suggest to include a few additional analyses and more methodological details, Reviewer 3 suggests that the current experimental evidence does not fully support the claims, because the detection task does currently not allow the authors to fully disentangle the contribution of movement preparation to SC responses. 

Given the extent of revision needed, we cannot make a decision about publication until we have seen the revised manuscript and your response to the reviewers' comments. Your revised manuscript is likely to be sent for further evaluation by all or a subset of the reviewers.

**IMPORTANT - SUBMITTING YOUR REVISION**

*Re-submission Checklist*

*Published Peer Review*

*PLOS Data Policy*

*Blot and Gel Data Policy*

Sincerely,

Christian

Christian Schnell, PhD

Senior Editor

PLOS Biology

cschnell@plos.org

REVIEWS:

Reviewer #1 (Michael Brecht): In their paper 'Stimulus selection drives value-modulated somatosensory processing in superior colliculus'Chu et al. record responses from tactile neurons in the mouse somatosensory cortex and superior colliculus during to go (lick reward) or nogo (nonreward) stimuli. The authors find largely symmetric go / nogo tactile responses in somatosensory cortex, but see a heavy bias to the reward predicting go stimulus in the colliculus. I think this is an interesting study. The strengths of the work are the following: (i) The simultaneous recording of two brain structures reveals important insights here. (ii) The somatosensory collicular representations are heavily understudied. (iii) The data are convincing. (iv) The effects are interesting and the reward bias in the colliculus makes biological sense.

My criticisms are minor:

1. The authors cite broadly and correctly most of the relevant literature, but missed two highly relevant references from the primate visual system.

Horwitz, G. D., & Newsome, W. T. (1999). Separate signals for target selection and movement specification in the superior colliculus. Science, 284(5417), 1158-1161.

Horwitz, G. D., & Newsome, W. T. (2001). Target selection for saccadic eye movements: direction-selective visual responses in the superior colliculus. Journal of Neurophysiology, 86(5), 2527-2542.

2. Individuals might differ in this task. In general, I think the data are convincing, but I would prefer to see two more S1 animals (i.e., a comparison of 6 and 6 micerather than of 6 and 4 mice). Just to be sure.

3. Did the authors ever poke into collicular tongue representation? See. Ito, Brendan S., et al. "A collicular map for touch-guided tongue control." bioRxiv (2024): 2024-04. This might provide further inside here.

Reviewer #2 (Saba Gharaei): In this study, the authors reveal how the representations of sensory space in S1 and SC are modulated by behavioural context. The authors recorded neuronal activity in S1 and SC of awake behaving mice performing a stimulus selection task that was guided by value-based associations. The authors found that stimulus value and its associated action did not modify the representation of sensory space in S1. However, the behavioural context influenced the representation of sensory space in SC where neuronal responses were biased towards the positive valued stimulus. 

When the opportunity to select the positive valued stimulus was removed, this bias in SC neurons was reduced. The experiment has an elegant design whereby the opportunity for mice to select the positive valued stimulus is removed with alternating trials in Home and Away conditions. Overall, this is a timely and well-performed study, the analyses seem appropriate, and the conclusions are supported by the neuronal and behavioural data.

I have only a few minor comments to enhance the clarity of the paper.

For the Hits vs Miss (or removing the opportunity to lick), the authors focused their analysis on neurons that preferred the positive stimulus. This is a logical choice. The authors demonstrate that for these SC neurons, the response to the positive stimulus was often larger on Hit trials compared to that of the Miss or Away trials. It is not clear to me whether the authors also looked at the activity of the same positively preferring neurons, to compare correct rejection trials and false alarms (licking to the no-go stimulus). It would be informative to see if the observed suppression of responses to the "no-go" stimulus is absent on trials that the animal made an error and licked to the no-go stimulus. 

More details of recording sites should be included. What was the depth of recordings? For S1, was the recording across all layers? For SC, was the recording only across the intermediate layers that are sensitive to somatosensory stimuli or across all layers of SC including the superficial and deep? It would be good to include this information given the observation that most SC neurons (76%, 549/727) displayed a significant tactile response, despite the fact that only 2 whiskers contacted the pole.

By comparing the peak responses in Fig 2B (value-based task) and 3C (equal value task), it is clear that the stimulus bias in SC, was due to the reduction of activity to the "no-go" stimulus rather than enhancement of responses to "Go" stimulus. Perhaps this could be made more explicit. Indeed, it was nice to see that this was further confirmed by the analyses in Fig 4. 

It would be useful to include further details about the behavioural methodology. How many days did it usually take for the training of mice for the whisker discrimination and the whisker detection task? 

In the method section, it would be useful to add where on the skull the head-bar was cemented to (i.e. describing Fig1D). As the same animals were used for both SC and S1 recordings (with SC being posterior and BC being anterior), it would be useful for readers to know more details of the custom-built head-bar (e.g. size). 

Reviewer #3: In this manuscript, the authors recorded single-unit activity in the superior colliculus (SC) and the somatosensory cortex (S1) while mice performed a go/no-go whisker discrimination task, allowing them to compare neural activity between the two brain areas during goal-directed behaviours. They found that the firing of SC neurons during whisker stimulation was biased towards the stimulus associated with reward (positive stimulus), while the S1 population had no such bias, consistent with the somatotopic map. Specifically, more SC neurons were positive stimulus-preferring, and their bias was stronger than in S1. To rule out the possibility that this result was driven by a sampling bias, the authors trained mice on a version of the task where both stimuli were associated with reward (detection task). In this task, the SC population showed little stimulus-bias in activity during touch. Further analysis of neural responses in the go/no-go task showed that the SC bias during touch was related to increased spiking on hit trials and delayed suppression of spiking on correct rejection (CR) trials in the positive stimulus-preferring population. Importantly, by comparing hit, miss, and away trials (i.e. where the waterspout was displaced so the animal couldn't respond), the authors found that positive stimulus preference in SC neurons is at least in part due to movement preparation. They also showed that baseline firing rates of SC neurons were lower during miss and away trials than engaged trials (hit + CR), and their baseline firing was correlated with response latency. From this, the authors concluded that SC somatosensory processing is modulated by value during stimulus selection, in contrast with the purely somatotopic mapping of S1.

The authors interpreted the bias in SC responses towards the positive stimulus as sensory encoding modulated by value. However, it is unclear if the bias was indeed related to sensory processing or was purely driven by movement preparation. The authors found that positive stimulus-preference in SC neurons was lower in no movement trials (miss and away), which is still consistent with the movement preparation interpretation. This seems plausible considering that the stimulus bias was quantified over a large time window from touch onset (multiple hundred milliseconds). This time window could include both early sensory and later movement preparation signals. The detection task used in this study did not allow the authors to fully disentangle the contribution of movement preparation to SC responses. Below we suggest some further analyses to address this issue. We recommend that the main conclusions of the paper, including the title and the abstract, be revised based on the results of these analyses. If no further evidence is found to support the SC value mapping interpretation, the conclusions should be adjusted to acknowledge the contribution of movement preparation. If these further analyses indeed support the current conclusions, this paper would provide novel insights into the role of SC in value processing for controlling actions.

1. The authors should analyse whether the performance difference between recording sessions (Fig. S1D) is reflected in the neural responses in SC (i.e. compare hit and CR responses in good and bad performance sessions between mice). If the positive stimulus bias in SC correlates with performance, this would support the value interpretation.

2. Neurons with early activity are likely to have sensory responses as opposed to movement preparation. The authors should quality stimulus bias in a shorter time window from first touch (like Fig. S3E but selecting neurons based on the early period only). Is the positive bias still enhanced in SC when only considering early responses (i.e. sensory neurons)? If yes, the main conclusion would be more supported.

3. From Fig. 5 the authors concluded that "sensorimotor processing in SC neurons is augmented by movement preparation", but it is possible that they are simply quantifying move preparation per se. They also later state that "positive preferring neurons may represent a specialized population of movement related neurons", which suggest they may agree with the movement preparation interpretation. The authors should also perform the analyses on the Away condition (e.g. Fig. 5E) on early sensory neurons to help clarify this, and they should make the text more consistent.

4. The authors should characterise neural dynamics on hit vs false alarm trials, where animals perform the same action to different stimuli. This could also help establish if the SC responses are value-modulated sensory or motor planning signals.

5. All neural data presented are aligned to first touch. Another approach to address whether SC neurons are sensory- or movement-related is to try different alignments of the neural data (i.e. first lick instead of first touch), and compare selectivity strength with different alignments (https://journals.physiology.org/doi/full/10.1152/jn.00508.2004).

Other suggestions:

1. The authors should provide more details on how they ensured that the sampling of neurons that respond to either whisker in the pair is balanced (like they already do for S1). Recordings from naïve animals could address this question.

2. The paper would benefit from more plots that show neural data subject-by-subject (e.g. for Fig 2C and E), especially because S1G suggests there is considerable variance in touch rates across mice.

3. The authors concluded from Fig. 4G-I that "touch suppression in SC may be computed from a different circuit than touch facilitation". That seems like a strong and not necessarily correct interpretation. Their interpretation in the Discussion, which offers both different circuits and local inhibition as possible mechanisms, seems more plausible.

4. The trial numbers in Fig. 5 and Fig. 6 seem to be different across contexts (hit/engaged, miss, away). The authors should match trial numbers (i.e. subsample from the hit/engaged and away states) to see if the conclusions still hold.

5. Related to Fig. 6, the authors state that "Baseline SC firing rates predict choice probability and speed". To make statements about choice probability, the authors would need to only compare miss and hit, rather than miss and engaged (hit + CR).

6. The clarity of the data presentation could be improved, especially the following panels:

a) It is confusing how Fig. 4D is different from Fig. 2B. The authors should explicitly state that the neurons in Fig. 4D are sub-selected based on baseline firing >4Hz and should provide a similar plot for the neurons <4Hz.

b) Can the authors make it clearer how the population of neurons in Fig. 4G relates to the ones in the other panels of Fig. 4?

Minor comments:

1. It would be helpful to show task performance (S1B-D) in the main figure.

2. The authors could statistically test whether SC has more positive encoding neurons compared to S1 (e.g. Chi squared test), rather than simply showing the percentage of neurons.

3. The authors could consider showing significance in the plots and not only in the text to make it easier for the reader to follow (e.g. Fig. 2C, E).

4. Typo in S3 title: "stimulus bias" instead of "stimulus".

---

## [Decision Letter · Decision Letter 2]

21 Jan 2025

Dear Scott,

Thank you for your patience while we considered your revised manuscript "Stimulus selection enhances value-modulated somatosensory processing in superior colliculus" for publication as a Research Article at PLOS Biology. This revised version of your manuscript has been evaluated by the PLOS Biology editors, the Academic Editor and two of the original reviewers.

Based on the reviews and on our Academic Editor's assessment of your revision, we are likely to accept this manuscript for publication, provided you satisfactorily address the following data and other policy-related requests:

* Please add the links to the funding agencies in the Financial Disclosure statement in the manuscript details.

* DATA POLICY:

Regardless of the method selected, please ensure that you provide the individual numerical values that underlie the summary data displayed in the following figure panels as they are essential for readers to assess your analysis and to reproduce it: 1DE, 2CE, 3DF, 4I, 5HI, S1FI, S2CDEF, S3DEF and S4BJK.

* Please ensure that you are using best practice for statistical reporting and data presentation. These are our guidelines https://journals.plos.org/plosbiology/s/best-practices-in-research-reporting#loc-statistical-reporting and a useful resource on data presentation https://journals.plos.org/plosbiology/article?id=10.1371/journal.pbio.1002128

* If you are reporting experiments where n ≤ 5, please plot each individual data point.

* CODE POLICY

We expect to receive your revised manuscript within two weeks. 

*Published Peer Review History*

*Press*

Sincerely,

Christian

Christian Schnell, PhD

Senior Editor

cschnell@plos.org

PLOS Biology

Reviewer remarks:

Reviewer #2 (Saba Gharaei): I am happy with the authors responses. 

Reviewer #3: The authors have addressed all of our concerns, and we support publication of this updated manuscript.

---

## [Editor Report · Decision Letter 3]

7 Feb 2025

Dear Scott,

Thank you for the submission of your revised Research Article "Stimulus selection enhances value-modulated somatosensory processing in the superior colliculus" for publication in PLOS Biology. On behalf of my colleagues and the Academic Editor, Mathew Diamond, I am pleased to say that we can in principle accept your manuscript for publication, provided you address any remaining formatting and reporting issues. These will be detailed in an email you should receive within 2-3 business days from our colleagues in the journal operations team; no action is required from you until then. Please note that we will not be able to formally accept your manuscript and schedule it for publication until you have completed any requested changes.

PRESS

Sincerely, 

Christian

Christian Schnell, PhD

Senior Editor

PLOS Biology

cschnell@plos.org